# OpenReview forum: "Are Large Language Models Really Robust to Word-Level Perturbations?"
_TMLR — Rejected by TMLR_

### Review · Reviewer_Ks8D · 2024-03-23

**Summary Of Contributions:**

This paper evaluates LLMs on their robustness to word-level perturbations that should not change the semantics of the input prompt and hence also should not degrade quality of the LLM outputs. The authors evaluate whether LLM performance degrades by evaluating generated LLM outputs with a reward model for helpfulness and harmlessness. They do this for 3 different type of perturbations: spelling mistakes, word swapping, and synonyms, and for 3 different LLMs of multiple sizes (Llama-2, Alpaca, and Beaver). The authors test LLMs using a subset of the Natural Questions dataset and find that the helpfulness of LLMs outputs are not robust to perturbations, but find no significant deterioration for harmlessness. Upon further investigation, the authors find that LLMs do get more harmful for perturbed responses after supervised fine-tuning.

**Audience:**

Yes

**Claims And Evidence:**

No

**Requested Changes:**

**Necessary for acceptance**

- Make sure the semantics of the input prompt are actually not changed by the perturbations. As mentioned above, now a more helpful response to most perturbed prompts would probably be something like "can you rephrase that question?" which might be much shorter than the response to the original prompt and therefore also receive different reward model scores.

- Justify your metric by showing it correlates with human judgements of helpfulness and harmlessness. This does not need to be done for all questions of course, but for a small subset of each type of perturbation and helpfulness and harmlessness.

- Make sure to control for the confounder of general bad performance of a model. A model can get perfect robustness by being maximally unhelpful and harmful on the original prompt, so the metric needs to somehow include original absolute performance on the clean prompt.

- Rewrite the parts of the paper where you claim evaluating generative LLM outputs with a reward model is a contribution of this paper to be more clearly about the actual contribution of using it for robustness to word-level perturbations.

**Minor requested changes**

- If you claim something is widely recognised, it needs citations. ("As widely recognized, landscape is frequently employed in ..." in section 4.3)

- The legend in figure 3 is very hard to read

- In many places in the text spaces are missing (couple times in abstract, in first paragraph of introduction, etc.). Almost always missing a space before a citation.

- The question on page 2 needs to be rewritten

- The paper is difficult to follow at times. For example, some of the conclusions from the results are difficult to follow because information is left out. It became clear to me after figure 5 that the Beaver model is fine-tuned from the Alpaca model, which is information you need earlier in the paper when you say that the Beaver family undergoes a notable deterioration in harmlessness robustness during SFT.

**Strengths And Weaknesses:**

**Strenghts**

- The authors address an important problem: robustness of LLMs to mistakes in the prompt. This is an important direction that can contribute to inclusiveness in evaluations.
- The authors test different model families of different sizes.
- The authors do a lot of experiments and evaluate both for harmlessness and helpfulness.

**Weaknesses**

- The perturbations the authors use do change the semantics of the input prompts, and nothing is mentioned about how the authors make sure the perturbations do not change meaning (e.g. swapping can easily change meaning). For most perturbed prompts I would personally not know how to answer (or at the very least have to simply guess what is meant). For example, the misspelled perturbation, quoting here without the original to make the point clear, *"what ss the mdaniiw of ueronica inu edgyish?"*. Another example where the perturbation makes the prompt ambiguous: "what is the meaning that veronica in English?". Of course you can guess what this prompt means, but my initial thought would be that something in this prompt is missing, and as an LLM I would respond "could you clarify what you mean here?" instead of responding with the meaning of Veronica in English. You claim that these perturbations occur frequently in user prompts for LLMs, but I see no evidence presented for this claim. This is a significant weakness of the paper, because it means that you are not actually measuring robustness to perturbations that maintain the original meaning of the prompt. If the meaning of the prompt changes, the reward model score give to the output is also not comparable to the reward model score of the original prompt (for a plethora of reasons but also because reward models are known to give higher helpfulness scores to longer outputs). In this case, a human evaluation of helpfulness between original and perturbed prompts can help (see also weakness below this one). With the perturbations you choose it's also not surprising that robustness decreases with more SFT and RLHF, because these fine-tuning techniques use high-quality data (without the spelling mistakes and perturbations you introduce), whereas pre-training data might be somewhat closer to the perturbations you introduce. This is mainly a weakness for evaluating helpfulness, as you still want a model to be harmless when these perturbations are used on prompts, so your metric can in a sense evaluate harmlessness robustness.

- When proposing a new evaluation method for human-centric things like "helpfulness" and "harmlessness", one needs to motivate this method by comparing it to existing metrics that measure the same, or if good ones do not exist, do a human study to see how it correlates with human judgement. This is not done in the paper, hence we need to assume that the reward model measures these things well and is not itself for example not robust. Additionally, you use the reward model from Beaver, which might give that model family an advantage over the other models. I would like to see another reward model being used and showing that the results presented here are robust to that.

- You measure robustness as the drop in reward score due to perturbations, but it could be that a model does bad on the original prompt, and equally bad on the perturbed prompts. How do you control for this confounder?

- Some of the conclusions are not supported by evidence from the paper. For example, you mention that "the higher robustness of Llama2-chat-13B compared to Llama-7B attests to the overall superiority of the Llama2 family who leverages additional resources and a more comprehensive training framework to ensure their performance", but your results show that Llama-7b is more robust than Llama-13b, which violates the above conclusion (meaning this conclusion can only be drawn for Llama-chat models). Additionally, unless Beaver-7B is indeed a model that is fine-tuned from the exact model Alpaca-7B (which depends on the model you used, but for example https://huggingface.co/PKU-Alignment/beaver-7b-v1.0 is fine-tuned from a reproduced Alpaca model), you cannot really compare the loss landscapes between Alpaca-7B and Beaver-7B and claim the difference is due to fine-tuning. Which exact models have you used for Alpaca-7B and Beaver-7B?

- Using reward models for evaluation of open-ended generative responses is not new (it is what methods using RLHF do and subsequently most LLM evaluation papers, for example an early work that does this is https://arxiv.org/abs/2009.01325). It's a stretch to claim this is a contribution of this paper. The paper is however written like that is a contribution (see point one of contributions starting on page 2 and the related work section paragraph on robustness evaluation of LLMs). Perhaps you are the first to use reward models to evaluate robustness to word-level perturbations, but in that case all the references to the contribution of using reward models for generated responses need to be caveated.

---

> ### Author Response · Authors · 2024-06-20
>
> Thank you for your thoughtful comments and time.
>
> Responses to the individual questions below.
>
> **[Question1]** The meaning of attacked prompts may be seriously misaligned with the clean prompts.
>
> Thanks a lot for this constructive point. We truly agree with the importance of maintaining semantic invariance of the original prompts. This reminds us that we need to introduce a metric to determine semantic invariance, and this metric needs to be aligned with human preference. Therefore, we create a small validation set from the evaluation dataset and introduce GPT-4 as a judge, who is widely used as a mirror of human preference[1][2]. We command GPT-4 to give opinions on whether the semantic of the sentence after perturbation differs from the original sentence. The prompts are show in **Appendix.A3**. The judgements by GPT-4 are shown below, the numerator represents if GPT-4 thinks the attacked prompt's semantic is different from the clean prompt. We can clearly see that misspelling is a more serious word level perturbation to LLMs. When the level reaches 3, the attacked prompts are absolutely different from the clean prompts and other type or level attacked prompts. We supplement more details of this validation set and evaluation prompts in **Appendix A.3**.
>
> Types&Levels  | 10% | 15% | 20% | 25% | 33%
> -------------------|------------------|------------------|------------------|------------------|------------------
> Misspelling  | 0/40  | 0/40 |  2/40 |  4/40 | 14/40
> Swapping| 0/40  | 2/40 |  1/40 | 1/40 | 2/40
> Synonym | 0/40  | 2/40 |  6/40 |  5/40 | 8/40
>
>
> **[Question2]** "helpfulness" and "harmlessness" needs to correlate with human judgement.
>
> Thanks for this advice! We sincerely agree with this constructive suggestion and introduce a new carefully trained reward model as judgement: ArmoRM-Llama3-8B-v0.1 [3] . This is a MoE reward model fine-tuned from llama-3-8B, and it ranks top_3 on the reward bench leaderboard [4], outperforming GPT-4-o from openai, gemini from google and so on. So its judgement is convincing, serving as a reliable supplement of the original Beaver reward model. We supplement an ablation study in **Appendix. A.2**, using this reward model and Alpagasus 9.2k Dataset to test the helpfulness robustness of Alpaca-7B-reproduce and Beaver-7B, the results show that Alpaca is more robust to word level perturbations than Beaver, although these two models both suffer from this attack. The ablation study is almost aligned with the results in our main context.
>
> Alpaca-7B-reproduce Average Drop Rate: 15.12%
> | Perturbation Level(%) | Level 1 | Level 1.5 | Level 2 | Level 2.5 | Level 3 |
> |-------|-------|-------|-------|-------|-------|
> | Misspelling     | 10.01|  21.02  | 28.82 |  38.85  | 52.92 |
> | Swapping       |  2.22 |   5.56  |  7.14 |   8.73  | 11.28 |
> | Synonym        |  2.20 |   6.03  |  7.78 |  10.31  | 14.01 |
> | Level Average Drop Rate | 4.81  | 10.87  | 14.58 |  19.30  | 26.07 |
>
>
> Beaver-7B Average Drop Rate  Beaver-7B: 15.79%
> | Perturbation Level(%) | Level 1 | Level 1.5 | Level 2 | Level 2.5 | Level 3 |
> |-----------------|-------|-------|-------|-------|-------|
> | Misspelling    |  10.38 |  21.74  | 29.46 |  39.66  | 53.30 |
> | Swapping       |    2.04 |   5.50  |  7.41 |   9.12  | 11.48 |
> | Synonym        |   2.20 |   5.98  |  7.86 |  10.52  | 14.15 |
> | Level Average Drop Rate |  4.87 |  11.07  | 14.91 |  21.76  | 26.31 |
>
> **[Question3]**  Is the tested Beaver model fine-tuned from Stanford Alpaca or PKU Alpaca-Reproduce?
>
> The Beaver model we use is a fine-tuned version from Alpaca-reproduce by PKU-Alignment team. We use them to make sure they have the same foundation, and one is fine-tuned from another.
>
> **[Question4]**  A model can get perfect robustness by being maximally unhelpful and harmful on the original prompt, so the metric needs to somehow include original absolute performance on the clean prompt.
>
> We give the absolute performance on the clean prompt in **Appendix A.4** by Beaver reward and cost model. We also supplement the results evaluated by ArmoRM-Llama3-8B-v0.1 on Alpagasus 9.2k Dataset in **Appendix A.2. Table 8** to solve this valuable concern.
>
> **[Question5]** Some writing problems occur in the article.
>
> Thanks for your careful review! We revise the writing problem, including missing space, typos, abbreviations definition, proper noun casual usage. Also, we supplement citations on necessary parts and rewrite the question on page 2 to make it more fluent and easy to understand.
>
> Reference:
>
> [1] https://github.com/tatsu-lab/alpaca_eval
>
> [2] Zheng, Lianmin, et al. "Judging llm-as-a-judge with mt-bench and chatbot arena." Advances in Neural Information Processing Systems 36 (2024).
>
> [3]https://huggingface.co/RLHFlow/ArmoRM-Llama3-8B-v0.1
>
> [4] https://huggingface.co/spaces/allenai/reward-bench

---

> > ### Comment · Reviewer_Ks8D · 2024-07-04
> > **Response to author rebuttal**
> >
> > Thanks to the authors for their detailed response and additional results! I will respond to everything below.
> >
> > - Response to Question 1: This essentially shows that for level 3 you cannot use this metric. I'd suggest removing all perturbed prompts for which the meaning has changed from the evaluation, not just for level 3, because if there's even 1 prompt for which the meaning has changed according to GPT-4 that could alter the reward scores. Now, it's unclear whether the reward score has dropped because of the changed meanings or because the model is not robust. It also seems like the decline in robustness due to fine-tuning you find is mostly explained by level 3, which could simply be the changed meanings. For level 1, the robustness actually gets mostly better / similar with fine-tuning.
> > - Response to Question 2: Why do you use a different dataset for this? I meant using a different reward model on exactly the same evaluation set as you use for the original questions, for the same score (so two reward models that evaluate helpfulness of the same 1k prompts). That way you can make sure the Beaver doesn't get an advantage over other models.
> > - Response to Question 3: Thanks!
> > - Response to Question 4: It's important to show these results in the main text in my opinion, because it's a confounder for your metric. Besides the drop rate in table 2 for example, also mention the absolute performance.
> > - Response to Question 5: Thanks!
> >
> > **Remaining and Additional concerns**
> > - The text still needs to be revised, perhaps the authors can use GPT-4 to help rewriting. Additionally, often still issues with missing spaces, uncapitalised words that should be capitalised, etc. Some examples that need revision just from page 2/3:
> > 	- "The former is often responsible for classification task, simple question and answer(Q&A) task, annotation task"
> > 	- "Therefore, there comes a question"
> > 	- "calculate their distinguish reward drop rates as an identification of robustness"
> > 	- "We find the LLMs suffer from word level perturbations." (here I'd add something like "the LLMs performance")
> > 	- "calling much attention on maintaining robustness while pursuing better model performance"
> >
> > - Contribution 1 still makes it sound like it's your contribution to evaluate on open-ended generation automatically.
> >
> > - Also a more irrelevant question but what does the T in TREvaL stand for?
> >
> > - I'm concerned about your method of selecting questions from Natural Questions. You select 1k questions for which the model generates the longest response. Given the length bias of most reward models, this means that you already have a biased test set here, and if you then perturb these prompts, the responses might get less long and not necessarily less good, but still get a lower reward. This is one of the reasons why I'd like to see a human eval. Even the new reward model you tried might have a length bias. I'd suggest using a random selection of 1k examples and not bias the set by selecting the longest response questions. Additionally, I suggest doing a small human evaluation experiment.
> >
> > - What does this mean: "We use NQ dataset to validate the robustness of the testing LLM by Beaver reward and cost model"?

---

> ### Author Response · Authors · 2024-07-06
>
> Thanks again for your thoughtful comments and time. We will answer every concerns you put forward. We try to supplement the experiments on a small amount.
>
> **[Concern 1]** The level3 is not suitable for evaluation, suggest using the prompts that do not change meaning. It’s unclear whether the reward scores drops due to robustness or changed meanings.
>
> Thanks a lot for this constructive point. First, we agree that changed meaning may somehow influence the evaluation of the robustness. Therefore we give a validation set, which shows the model’s robustness under the premise of how much the current disturbance changes prompt semantics. We will follow your advice and show this validation results before giving the models’ robustness drop results. It can serve as a priori condition before seeing our results and conclusions.
>
> Second, to solve this concern, we conduct an experiment:
> 1. we first use GPT-4 to judge the whole prompt dataset(Natural Questions dataset) of misspelling as an example, and save the prompts that do not change semantics after the perturbation.
>
> 2. Then, we randomly select 1000 prompts and their corresponding attacked prompts, we conduct this on level 1 ,2, 3.
>
> 3. We show the results below:
>
> Models&Levels  | 10% | 20% | 33%
> -------------------|------------------|------------------|------------------
> Alpaca  | 6.72  | 9.84|  15.32
> Beaver| 9.72 | 18.00|  23.59
>
> The results show again that the robustness drops in fine-tuning process. Also, our loss-landscape shows the decline of robustness in the fine-tuning process, as well as the results on ArmoRM-Llama3-8B-v0.1, serving as another strong proof.
>
> **[Concern 2]** Why a new dataset?
>
> We introduce a new dataset due to the concern of monotony evaluation . We notice that only one datasets may be not convincing enough for evaluation, so we introduce another instruction following dataset.
>
>
> **[Concern 3]** Please show the absolute performance in the main context.
>
> We will move this part to the main context for better reading experience.
>
> **[Concern 4]** The selection method from NQ may hacking reward model’s judgment.
>
> Thanks for this concern. We agree that the reward model tend to prefer longer responses. However, as we use the same prompts(for the tested model, the prompts before and after the perturbs are nearly the same), we think this variable is within a controllable range.
>
> **[Concern 5]** Details about the article.
>
> Thanks for your careful review, your suggestions are really important for us to make this work more readable. We will revise the missing spaces, uncapitalised words to achieve a more fluent expression.

---

> > ### Comment · Reviewer_Ks8D · 2024-07-07
> > **Thanks for these responses**
> >
> > Thanks a lot again to the authors for responding so swiftly and updating with new results.
> >
> > **Changed meanings**
> >
> > This indeed shows that there is an issue with robustness to misspellings, even if the semantics is unchanged (most likely, I still believe a small human study on these beyond GPT-4 would be useful, but it's not a necessity). Given the large difference in results when you do not filter out the prompts where the meaning has changed (the robustness drop is much smaller), I think it's important to do this for the whole dataset of prompts and not just misspellings on a small subset.
> >
> > **New dataset**
> >
> > Thank you for that, that is indeed also useful. However, my original concern was about making the evaluation of models more robust by using multiple reward models to evaluate a single model on a single dataset.
> >
> > **Absolute scores**
> >
> > Great!
> >
> > **Method of selecting questions**
> >
> > I respectfully disagree. The prompts might be the same, but you selected based on the output of the model for the original prompt (unperturbed). This means that you are selecting with a bias towards higher reward for your baseline evaluation against which you measure the drop in reward, which will bias the results. I believe it's important to randomly select a set of prompts and not the longest. I also am not sure what the benefit is of selecting the longest prompts.

---

> ### Author Response · Authors · 2024-07-07
>
> We sincerely appreciate your meticulous reading of our work and the many constructive suggestions you have provided. We truly think that this revised article with your suggestions is becoming better and better. Your careful review plays a unique role!
>
> **[Concern 1]** Changed meaning
>
> We notice the rebuttal period is drawing to an end, and that's why we could only conduct a small number of experiments. We will further follow your advice to conduct ablations on the entire dataset of prompts and not just misspellings on a small subset!
>
> **[Concern 2]** New dataset
>
> We sincerely agree with this concern, that the results of a new reward model on the same prompt set is much more convincing. We will soon follow this advice and implement this ablation study!
>
> **[Concern 3]** Method of selecting questions
>
> Thanks for your careful and professional review! We will explain why we use the length as a measurement.
>
> 1. We agree with that reward model tends to give higher rewards to longer responses.
> 2. Actually, we initially used this filtering method only to select those more open questions. We intend to avoid questions that the model will answer with only one or two words, such as classification questions or questions having a clear answer. For example, for the question: "Who is the president of America in 2018?", the ground truth label could be "Donald trump". Answering this will get a perfect reward with a classifier or a ground truth label. This is also what some other evaluation metrics conduct[1][2]. However, as we mention in **Introduction**, we aim to make full use of the generative capacity of LLM and induce the language models to generate extensive responses. We would like to see if the robustness of the LLMs tend to drop with these open questions and that's a big difference between us and other evaluation metrics. We are investigating if LLMs tend to speak shorter and speak worse.
>
> Reference:
> [1]Instructeval: Towards holistic evaluation of instruction-tuned large language models[J]. arXiv preprint arXiv:2306.04757, 2023.
>
> [2]Promptbench: Towards evaluating the robustness of large language models on adversarial prompts[J]. arXiv preprint arXiv:2306.04528, 2023.

---

> > ### Comment · Reviewer_Ks8D · 2024-07-07
> > **Response**
> >
> > Thank you for engaging so actively. I agree that running the full experiments is out of the scope of the rebuttal period. Additionally, I understand now why you only select the longest outputs. Perhaps there is a less stark way you can select open questions. E.g. more than X amount of words, but not just the longest bucket.
> >
> > To summarise, I believe this paper has the potential to present an interesting and insightful result, but believe it necessary to:
> > (1) perform the full experimental protocol on the prompts without changed meaning
> > (2) make sure there is no strong length bias by only selecting the longest prompts for evaluation.
> > I suspect that even after applying this, the models will still be not perfectly robust to perturbations.

---

> ### Author Response · Authors · 2024-07-08
>
> Thanks for your suggestions! We notice that although the rebuttal period is end, we can still comment on this page. So we will try our best to conduct the experiments.
> We sincerely appreciate your advice, and will follow it in detail. We keep the original prompts, but indicate how we select them to achieve model's generative capacity. We will use GPT-4 to keep the semantics of the prompts, and do not change their meaning.
>
> Thanks again for your patience. And now we give the revised performance :
>
> 1. We first give the performance after filtering the changed prompts with GPT-4 :
>
> (1)Test model:Alpaca-7b-reproduce, Reward model: Beaver-Reward model, Average Drop rate: 7.39%
>
> Types&Levels(%)  | 10% |  20% |  33%
> -------------------|------------------|------------------|------------------
> Misspelling  | 6.72  | 14.41 |  24.98 |
> Swapping| 0.54  | 4.54|  7.51 |
> Synonym | 0.27  | 2.74 | 4.76 |
>
>
> (2)Test model:Beaver-7b, Reward model: Beaver-Reward model, Average Drop rate: 9.25%
>
> Types&Levels(%)  | 10% |  20% |  33%
> -------------------|------------------|------------------|------------------
> Misspelling  | 9.72  | 16.50 |  29.44 |
> Swapping| 2.08  | 5.72|  10.02|
> Synonym | 0.69  | 3.41 | 5.56 |
>
> This results is in line with our conclusion, and we summarize them into our work.
>
> 2. We conduct experiments on the same prompts with another reward model: ArmoRM-Llama3-8B-v0.1
>
> (1)Test model:Alpaca-7b-reproduce, Reward model: ArmoRM-Llama3-8B-v0.1, Average Drop rate: 20.76%
>
> Types&Levels(%)  | 10% |  20% |  33%
> -------------------|------------------|------------------|------------------
> Misspelling  | 20.19  | 42.89 |  61.54 |
> Swapping| 5.98  | 11.19 | 17.28  |
> Synonym | 3.52  | 7.82 | 16.40 |
>
> (2)Test model:Beaver-7b, Reward model: ArmoRM-Llama3-8B-v0.1, Average Drop rate: 21.78%
>
> Types&Levels(%)  | 10% |  20% |  33%
> -------------------|------------------|------------------|------------------
> Misspelling  | 21.95  | 43.91 |  62.27 |
> Swapping| 6.73  | 12.15|  18.80|
> Synonym | 4.83  | 8.68 | 16.62 |
>
> We implement these ablation studies as you suggest, and we actually think these new experiments will make our article much convincing.

---

> > ### Comment · Reviewer_Ks8D · 2024-07-08
> >
> > Thanks for these additional results, I agree they much improve the paper. The only remaining concerns are then the selecting method of the questions (how to deal with not selecting multi-choice / non-open-ended questions), and a potential small human study (especially since the different reward models give such different results!). However, as these changes are quite substantial, I  do believe the paper can benefit from another round of reviews, as I'd need to judge the revision anew.

---

### Review · Reviewer_xjEC · 2024-06-04

**Summary Of Contributions:**

This paper proposes a pipeline to evaluate the robustness of LLM on the QA task after adding some word-level perturbations. The paper tests the proposed pipeline on different perturbation method and  perturbation level across several LLMs at different stages, leading to a solid evaluation of the proposed pipeline.

**Audience:**

Yes

**Claims And Evidence:**

No

**Requested Changes:**

Please see above weaknesses.

**Strengths And Weaknesses:**

Strengths:

The experiments done in this paper is solid, making the claims and conclusions persuasive.

Weaknesses:

I don't see big issues of this paper, but I do see some minor problems needed to be addressed:

1. **It's obvious that this paper has been revised through ChatGPT, but the issue is some parts lose necessary logic, making the paper hard to understand.** For example, in Introduction, "Although it is reasonable to use traditional NLP tasks to test the robustness of pretrained LLMs, it is also worth considering whether a traditional Bert-based task can fully identify the capacity of a LLM". The later content seems weakly correlating with the mentioned "it is also worth considering whether a traditional Bert-based task can fully identify the capacity of a LLM".

2. **The perturbation methods has improvement space**. For the currently used word-level perturbation, the perturbation added to keywords or unimportant words like conjunctions will definitely lead to different model performance. The author should discuss this. Additionally, why the author didn't consider adding/deleting words?

3. **The paper loses some details.** For example, why the authors only choose 1k open questions from the available dataset instead of using all of them? What are the selection criteria?

---

> ### Author Response · Authors · 2024-06-19
>
> Thank you for your thoughtful comments and time.
>
> Responses to the individual questions below.
>
> **[Question1]** The writing needs improvement.
>
> Thanks for your careful review, we revise the writing problem in this article, including missing space, typos, abbreviations definition, proper noun casual usage, and we will strive to perfect every detail.
>
> **[Question2]** The perturbation method has improvement space.
>
> Thanks for your suggestion, we choose the three types of perturbation as examples of common misusage. We indeed agree with that the perturbation added to keywords or unimportant words like conjunctions will definitely lead to different model performance, so we include GPT-4 as a judge and introduce a small validation set as supplement in **Appendix A.3 Table 9**, to measure if the perturbation types and levels added to the prompts lead to significant misunderstanding. Specifically, we use a validation size of 40 prompts and ask GPT-4 to judge if any perturbation type or level conducted in this study leads to serious semantics deviation. GPT-4 is widely used to evaluate performance of LLMs[1][2], and we choose a size of 40 prompts due to the expensive Openai Api cost. The results are listed as following shows,  the numerator represents if GPT-4 think the attacked prompt’s semantic is different from the clean prompt:
>
> Types&Levels  | 10% | 15% | 20% | 25% | 33%
> -------------------|------------------|------------------|------------------|------------------|------------------
> Misspelling  | 0/40  | 0/40 |  2/40 |  4/40 | 14/40
> Swapping| 0/40  | 2/40 |  1/40 | 1/40 | 2/40
> Synonym | 0/40  | 2/40 |  6/40 |  5/40 | 8/40
>
> **[Question3]** The paper lost some details such as dataset selection.
>
> Thanks for this careful review. We indeed agree with the necessity to explain dataset selection in detail. The datasets are selected via their generated responses' average length. Specifically, we select the longest 1k average responses via an instruction following LLM, LLaMA-2-chat-7B. For each prompt, we generate two responses and calculate the average length. We select the top 1k prompts. We supplement this explanation in our article in **Section 3.1**.
>
> Reference:
>
> [1] https://github.com/tatsu-lab/alpaca_eval
>
> [2] Zheng, Lianmin, et al. "Judging llm-as-a-judge with mt-bench and chatbot arena." Advances in Neural Information Processing Systems 36 (2024).

---

### Review · Reviewer_C6ML · 2024-06-08

**Summary Of Contributions:**

This work aims to evaluate the robustness of large language models (LLMs) in view of their generation ability. Different from previous studies that rely on pre-defined supervised labels, this work proposes a novel rational evaluation pipeline (TREvaL) that leverages reward models as diagnostic tools to evaluate the long conversation generated by LLMs. Given the questions, word-level perturbations are adopted. The chosen perturbation methods include synonym substitution, swapping, and misspelling. Specifically, three levels of perturbation are employed. For evaluation, this work leverages the questions from Natural Questions (NQ) dataset and evaluates the generated content by a reward model. Empirical experiments are based on Llama, Alpaca, Beaver, and Llama 2 models. The results demonstrate that the proposed TREvaL provides an identification for the lack of robustness of nowadays LLMs. Notably, robustness tends to decrease as fine-tuning (SFT and RLHF) is conducted, calling for more attention on the robustness during alignment process.

**Audience:**

Yes

**Claims And Evidence:**

Yes

**Requested Changes:**

1. Experimental settings and choices need clarification.

- The reward model is a critical part in this work. However, there is only a short reference of the Beaver-7B Reward Model. It is not clear what exactly the reward model is.

- It is not clear how the evaluation scores are calculated. In Section 4.2.1, helpfulness robustness gap and harmlessness robustness gap scores are calculated. As this is a generation problem, it is not clear how the scores are obtained given the generated content. I believe that there might be an scoring function. But I could not find how the function is implemented (e.g., by which model), and which metric is used.

- The choice of the evaluation dataset is not justified. This work only use a traditional NQ dataset. It is doubtful that why the dataset is suitable for this task and why not conduct more comprehensive experiments on more widely adopted datasets for LLM evaluation to show the generality.

2. The writing needs significant improvements.

- There are many places missing spaces between the text and punctuation. Here are some examples:
"words or letters.Our extensive" -> "words or letters. Our extensive"
"robustness of nowadays LLMs.Notably, we are surprised" -> "robustness of nowadays LLMs. Notably, we are surprised"
"embodied agent tasks(Di Palo et al., 2023; Huang et al., 2023a; Li et al., 2023a)." -> "embodied agent tasks (Di Palo et al., 2023; Huang et al., 2023a; Li et al., 2023a)."
"choose 1k prompts(open questions)" -> "choose 1k prompts (open questions)"

- Typos
"Bert" -> "BERT"
"a LLM" -> "an LLM"

- Some descriptions are hard to understand.
" As the dataset is a mixup of open-ended and open questions, we try to avoid the open-ended questions and choose 1k prompts(open questions) from a 5.6k set to best leverage the generative capabilities of LLMs"
So did you use 1k open questions or the rest for the evaluation?

- Some abbreviations are not defined before use, e.g., "SFT", "RLHF".
- Some terms are used in chaos. There are "QA" and "Q&A" used casually.

**Strengths And Weaknesses:**

1. This work studies an interesting problem by evaluating the robustness of LLMs based on the generated content instead of pre-defined supervised label. This setting adapts to the era of the generative LLMs.

2. Various word-level perturbation methods, i.e.,  synonym substitution, swapping, and misspelling, under three levels, are adopted. Three types of LLMs are evaluated. The experimental results are generally in line with intuition.

---

> ### Author Response · Authors · 2024-06-19
>
> Thank you for your thoughtful comments and time.
>
> Responses to the individual questions below.
>
> **[Question1]** The reward model is a critical part in this work. However, there is only a short reference of the Beaver-7B Reward Model. It is not clear what exactly the reward model is.
>
> Thanks for this question, we agree that a more detailed description of the reward model is necessary. Therefore, we follow your advice and elaborate on the usage of the Beaver reward model and cost model in **Section 3.2**. Moreover, to make our experiments more compelling, we supplement a new reward model: ArmoR-Llama 3-8B-v0.1 and conducted experiments on a novel open question dataset, the Alpagasus 9.2k Dataset on Alpaca-7B and Beaver-7B models in **Appendix A.2**.
>
> **[Question2]**  It is not clear how the evaluation scores are calculated.
>
> Specifically, a reward model scores a set of Q&A pairs before and after word level disturbances. Notably, we observe that the original reward scores are decimals between -20 and 20. So we try to normalize the scores from 0 to 100 and calculated the rate of decline for each normalized score, reporting it as the “drop rate”. We notice the absence of explanation on scores metric may mislead the author, so we supplement the details in **Section 3.2 "Method”** paragraph.
>
> **[Question3]** The choice of the evaluation dataset is not justified.
>
> Thanks for pointing out this issue, we initially decided to use the open question to do the calculation. In the Natural Question Dataset, there are not only open-question, but also some questions that have a clear answer in a single word, such as "what does the word china mean in chinese?" refers to "中国". So we use an instruction following model to twice generate responses and select the longest 1k responses' prompts. Also, we indeed find that only use the NQ dataset is not convincing enough, so we introduce a new open question dataset: Alpagasus 9.2k Dataset as a supplement and report performance on it in **Appendix. A.2**.
>
> **[Question4]** There are many places missing spaces between the text and punctuation, the writing needs improvement.
>
> Thanks for your careful review, we revise the writing problems in this article, including missing space, typos, abbreviations definition, proper noun casual usage. We hope the revised version of our article can bring you a better reading experience.

---

### Author Response · Authors · 2024-06-19

Dear Reviewers,

We are extremely grateful for your valuable feedback and insightful comments. Your concrete suggestions are a valuable step in this direction, and we have revised our submission accordingly to take these into account. In this short note we summarize the main changes in that test revision of our submission:

1. Add a more clear reward model clarification in **Section 3.1**.

2. Introduce a new reward model and a new open question dataset to ensure diversity and preference consistency with humans in **Section 3.1**, and report the results of the new experiments as a supplement  in **Appendix A.2**.

3. Add detailed explanation on how the reward scores are calculated in **Section 3.2**.

4. Show the absolute performance on the clean prompt in **Appendix A.2** and **Appendix A.4**.

5. Revise the writing problem in the article, including missing space, missing citations, typos, abbreviations definition, proper noun casual usage.

6. Give a validation set to detect the semantic consistency with human preference before and after word level perturbations in **Section 3.3** "Method" and report the results in  **Appendix A.3**.

---

### Decision · Action_Editor_FkaR · 2024-07-17

**Recommendation:** Reject

**Comment:**

All the reviewers acknowledged the strengths of the paper, including that the proposed TREvaL pipeline is innovative in using reward models to evaluate LLM robustness through long conversations generated from challenging open questions, and the various word-level perturbations (synonym substitution, swapping, and misspelling) across multiple LLMs (Llama, Alpaca, Beaver, Llama 2) and different perturbation levels is very comprehensive.

However, after discussion, the reviewers are still leaning to reject this paper due to the following reasons:
1) Experimental Setting: Reviewer Ks8D points out that the experimental protocol is not sound. The main issue is that perturbed prompts often change the meaning of the original prompts, which confounds the evaluation of robustness. The selection of the longest responses for evaluation introduces a bias. This could skew the results since longer responses are more likely to receive higher reward scores.
2) Reward Model Justification: There is a need to justify the use of reward models by showing a correlation with human judgments. The current approach relies heavily on the Beaver reward model without sufficient justification.

Therefore, this paper is not yet ready for TMLR.

**Audience:**

general interest

**Claims And Evidence:**

The paper investigates the robustness of large language models (LLMs) using a novel evaluation pipeline (TREvaL) that leverages reward models to assess long conversations generated by LLMs from challenging open questions. The study found that LLM robustness decreases with fine-tuning (SFT and RLHF). Reviewers appreciate the improvements but identify major flaws in the experimental protocol and suggest substantial revisions, including addressing changes in prompt meaning, evaluating different reward models, and ensuring unbiased prompt selection. The authors responded with detailed revisions, new datasets, improved writing, and additional validation, but reviewers maintain concerns, suggesting further substantial revisions and thus not suitable for publication.

**Resubmission Of Major Revision:**

The authors may consider submitting a major revision at a later time.